# Intake of Special Amino Acids Mixture Leads to Blunted Murine Colon Cancer Growth In Vitro and In Vivo

**DOI:** 10.3390/cells13141210

**Published:** 2024-07-18

**Authors:** Giovanni Corsetti, Claudia Romano, Silvia Codenotti, Lorena Giugno, Evasio Pasini, Alessandro Fanzani, Tiziano Scarabelli, Francesco S. Dioguardi

**Affiliations:** 1Department of Clinical and Experimental Sciences, University of Brescia, 25123 Brescia, Italy; cla300482@gmail.com (C.R.); lorena.giugno@unibs.it (L.G.); evpasini@gmail.com (E.P.); 2Department of Molecular and Translational Medicine, University of Brescia, 25123 Brescia, Italy; silvia.codenotti@unibs.it (S.C.); alessandro.fanzani@unibs.it (A.F.); 3Italian Association of Functional Medicine, 20855 Lesmo, Italy; 4Holy Cross Medical Center, Taos, NM 87571, USA; tscarabelli@hotmail.com; 5NutriResearch Srl, 20127 Milano, Italy; fsdioguardi@gmail.com

**Keywords:** cancer, amino acids, colon, nitrogen, diet, mice

## Abstract

Cancer cells require substantial amounts of energy and substrates for their metabolic hyperactivity, enabling the synthesis of new cells at the expense of healthy ones. Preliminary in vitro data suggest that a mix of free essential amino acids (EAA-mix) can promote cancer cell apoptosis by enhancing autophagy. This study aimed to confirm, both in vitro and in vivo, whether EAA intake could influence the development of colon cancer in mice. We investigated changes in cancer proliferation in CT26 cells treated with EAA-mix and in mice fed with EAA-rich modified diets (EAARD) as compared to those on a standard laboratory diet (StD). CT26 cells were injected subcutaneously (s.c.) or intraperitoneally (i.p.). After 21 days, tumors were removed and measured. In vitro data corroborated that EAA-mix impairs cancer growth by inducing apoptosis. In vivo data revealed that mice on StD developed significantly larger (s.c.) and more numerous (i.p.) cancers than those on EAARD. EAA administration appears to influence cancer cell survival with notable antiproliferative properties.

## 1. Introduction

Colon cancer ranks globally as the third most common cancer type and the third-leading cause of cancer death. It has an estimated 5-year relative survival rate of 64% [1]. Hence, any intervention that improves survival rates for colon cancer patients can have significant implications.

Tumor growth, driven by the duplication of abnormal cells, requires high biosynthetic activity. This, in turn, demands an adequate supply of both nutrients and energy. We have postulated a possible energy-dependent relationship balancing the high-energy consuming process of syntheses and autophagy (AUT) [2]. Protein synthesis consumes a large amount of ATP, generating AMP that subsequently activates AMP kinase (AMPK). AMPK activation would then suppress mammalian target of rapamycin components (mTORCs)-dependent syntheses and trigger the AUT machinery, promoting ATP and nonessential amino acids refueling in a continuous synchrony [2,3].

Amino acids (AA) are broadly categorized into essential and nonessential types. Despite some constraints, this classification is relatively uncomplicated to comprehend and implement. Essential amino acids (EAA) necessitate consistent replenishment via dietary intake since mammals possess limited capacity for their autonomous synthesis, a deficiency that is life-incompatible. Conversely, nonessential amino acids (NEAA) are bountifully supplied by food, yet the body can also produce them as required, using EAA as starting points. It is widely recognized that the presence of all EAA in adequate quantities is the limiting determinant in protein synthesis.

Tumor cells exhibit profound alterations in amino acid (AA) metabolism, which sustains protein anabolism and energy production for proliferation support. As a result, colon cancer cells release diffusible molecules in the bloodstream that stimulate protein and lipid depletion, mainly in the skeletal muscle, giving rise to cachexia. Therefore, EAA supplementation may improve the nutritional status of cancer patients by counteracting cachexia and enhancing the effectiveness of chemotherapy treatments by minimizing collateral damage [4].

We have previously explored the safety and benefits of food supplementation with a specific free AA mixture rich in EAA (EAA-mix) under various experimental conditions [5,6,7,8,9,10,11,12,13,14]. Some reports have discussed how reducing certain NEAA (serine and glycine) and consequently increasing EAA may impact cancer in animal settings [15,16]. However, it was not openly acknowledged in Methods that, by subtracting some NEAA such as serine and glycine from tested formulations, EAA percentages would naturally increase [17], as discussed elsewhere [18].

We have previously demonstrated that altering the EAA/NEAA ratios in diets significantly influences mice survival [19]. Moreover, we have shown in vitro that an EAA-mix containing all free EAA in a stoichiometric ratio inhibits tumor cell proliferation and survival by inducing both AUT and apoptosis in duplicating cancer cells [7,20]. Therefore, we investigated whether changes in EAA/NEAA ratios in the diet also affect in vivo cancer development.

Given that the EAA-mix of the special diet also contained serine for specific biological reasons, this study challenges previous data and opinions about the role of EAA and serine in cancer development [16,18].

We initially conducted an in vitro study and subsequently an in vivo study, both via subcutaneous (s.c.) and intraperitoneal (i.p.) injection, using murine colon tumor cells (CT26). In the in vitro study, our primary objective was to assess the effects of the EAA-mix in inducing the death of CT26 cells. In the in vivo studies, our primary objective was to evaluate the difference in tumor development (final volume) between animals fed a standard laboratory diet (StD) and those fed a special EAA-rich diet (EAARD) as the exclusive nitrogen source.

## 2. Materials and Methods

### 2.1. In Vitro Experiments

Cell Cultures. The mouse CT26 colon carcinoma cell line (kindly provided by Prof. Paola Costelli and Prof. Fabio Penna) was cultured in a humidified incubator at 37 °C with 5% CO_2_. The cells were maintained in high-glucose DMEM (Euroclone, Milan, Italy) supplemented with 100 mg/mL penicillin/streptomycin (Sigma Aldrich, Milan, Italy) and 10% FBS (Euroclone, Milan, Italy).

Preparation of AA Solution. Cells were treated with a solution of EAA-mix or NEAA-mix (Table 1) dissolved in complete medium (DMEM supplemented with 100 mg/mL penicillin/streptomycin and 10% FBS). AA were dissolved at 1% (*w*/*v*) concentration in the medium; then, the solution was vortexed and sonicated at 50 °C for 30 min to reach complete solubility. Cells were incubated with medium containing 1% or 0.5% AA-mix concentration. Control cells were treated with complete growth medium.

Neutral Red Assay. Cell viability was measured by neutral red assay. Cells (2 × 10^3^) were seeded in triplicate in 96-well plates and left to grow in complete medium for 24 h. Then, cells were incubated with a medium supplemented with EAA-mix or NEAA-mix at 0.5 or 1% concentration. After 48 and 72 h, medium was replaced with DMEM supplemented with 5% FBS and 40 μg/mL neutral red dye and plates were incubated at 37 °C for 2 h. Then, cells were PBS-washed and incubated with a destaining solution (50% ethanol in deionized water with 1% acetic glacial acid). Plates were shaken until complete dye extraction was achieved and then absorbance was measured by reading the plate at 540 nm emission wavelengths.

Flow Cytometric Analysis. Cell apoptosis was assessed using an Annexin V/Propidium Iodide (PI) apoptosis-detection kit (ImmunostepBiotec, Salamanca, Spain), according to the manufacturer’s instructions. Cells (5 × 10^4^) were seeded in duplicate into 6-well plates and left to grow in complete medium for 24 h. Then, cells were incubated with a medium supplemented with EAA at 1% concentration. After 48, 72, and 96 h, cells were collected into flow cytometry tubes, PBS washed, resuspended in binding buffer, and double stained with Annexin-V-FITC/PI. Doxorubicin was used as a positive control. Cytofluorimetric analysis was performed using a MACSQuant Analyzer (Miltenyi Biotec, Bologna, Italy). Cell debris, doublets, and aggregates were excluded from the analysis, and 20,000 events per sample were analyzed.

Immunofluorescence Analysis. Cells were cultured onto 12 mm glass coverslips in 24-well plates (2 × 10^4^) and left to grow in complete medium for 24 h. Then, cells were incubated with a medium supplemented with EAA-mix or NEAA at 0.5 or 1% concentration. After 24 and 48 h, cells were fixed with paraformaldehyde (PFA) solution (3% PFA in PBS) for 20 min at 4 °C, permeabilized with 0.1% Triton X-100 in PBS for 10 min at room temperature (RT), and then blocked with a bovine serum albumin (BSA) solution (1% BSA in PBS with 0.1% sodium azide) for 30 min at RT. Cells were incubated with primary antibody anti-LC3β (code: sc-376404; Santa Cruz Biotechnology, Dallas, TX, USA) for 3 h at RT, washed, and then incubated with secondary antibody for 45 min at RT, protected from light. Nuclei were counterstained with Hoechst dye for 30 sec at RT and samples were mounted on slides using Mowiol mounting media. Images were acquired by using a fluorescent Axio observer microscope equipped with Apotome (Carl Zeiss, Oberkochen, Germany) using Zen 3.5 (Blue Version) software (Carl Zeiss, Oberkochen, Germany). 

### 2.2. In Vivo Experiments

The experimental protocol was approved and conducted in accordance with laws of the Italian Ministry of Health and complied with the “The National Animal Protection Guidelines”. The Ethical Committee for animal experiments of the University of Brescia (OPBA) and the Italian Ministry of Health had approved the procedures (decree n. 539/2021-PR).

### 2.3. Diets

Standard laboratory rodent food (Mucedola srl, Milan, Italy) was used as the reference standard diet (StD). The special EAA-mix-rich diet (EAARD) (Dottori Piccioni s.r.l., Gessate, Milano, Italy) matched the same total macronutrients, micronutrients, and calorie contents. The primary difference between the diets was the source and type of nitrogen. In StD, the nitrogen source was represented by unspecified vegetal and animal (fish) proteins. As with all food proteins, the EAA-to-NEAA ratio (EAA/NEAA) should be considered < or <<0.9. From whole protein mix, it is impossible to obtain the exact percentage in EAA and NEAA. In contrast, EAARD provided nitrogen as free AA, where the EAA are in excess (84%) compared to NEAA (16%) (EAA/NEAA = 6.14), as previously described [19]. In summary, both the EAA-mix and the EAARD incorporated L-cystine (NEAA) to satisfy the requirements for sulfur AAs while keeping methionine content to a minimum. Additionally, serine (NEAA) was included to optimally sustain the folate and methionine cycle, as well as energy production.

The composition of the EAA-mix and pellets is summarized in Table 1.

Animals. Fifty BALB/c male mice, aged 5 weeks (Envigo srl, San Pietro al Natisone, Udine, Italy), were individually housed in filter cages and kept on a 12/12 h light/dark cycle. After 7 days of ad libitum access to StD and water, the mice were randomized into two groups. The first group (n = 25) continued with the StD, while the second group (n = 25) was switched to the EAARD. Every three days, body weight (b.w.), food, and water consumption were measured.

After 15 days, 10 StD-fed mice and 10 EAARD-fed mice were injected subcutaneously (s.c.) at the right hip with 1 × 10^5^ CT26 cells (ATCC code CRL-2638™) strain Balb/c in 100 μL of physiological solution. An additional 30 mice (15 StD- and 15 EAARD-fed) were intraperitoneally (i.p.) injected with 1 × 10^6^ CT26 cells.

Daily monitoring was performed and, if necessary, animals were euthanized early based on veterinary advice and Ethics Committee Criteria. At 21 days post-injection, all injected animals were sacrificed. S.c. tumors, rpWAT, BAT, and triceps of surae were isolated, measured, and weighed. Similarly, visible i.p. tumors with a diameter greater than 1 mm located on the mesentery and/or on the parietal peritoneum were measured and weighed. Tumor volume (mm^3^) was calculated as:Vol = [(max diameter × smallest diameter^2^)/2]

All tumors were stored appropriately for histological and immunohistochemical analysis.

Histology and Immunohistochemistry. Tumor samples were embedded in paraffin using an automatic includer (Donatello-2, Diapath s.p.a, Martinengo, BG, Italy). Histopathological analysis was conducted under eosin and hematoxylin (E/H) staining [21,22]. Collagen production was evaluated with a picrosirius stain (Sirius-red), as described [23,24,25].

Tumor sections were incubated overnight with primary polyclonal anti-Ki-67 (28074-1-AP), anti-CD31 (28083-1-AP), anti-GRP78 (11587-1-AP), recombinant anti-iNOS (80517-1-RR), all from Proteintech (Rosemont, IL, USA), and anti-active Caspase-3 (NB100-56113) from Novus Biologicals (Easter Ave Centennial, CO, USA). The sections were processed according to the manufacturer’s protocol and visualized with IHC Prep & Detect Kit for Rabbit Primary Antibody (PK10017, Proteintech). The IHC negative control was performed by omitting the primary antibody in the presence of iso-type-matched IgGs. The staining intensity was evaluated using an Olympus BX50 microscope equipped with an image analysis program (Image Pro-Plus 4.5.1, Immagini e Computer, Milano, Italy). The IOD was calculated for arbitrary areas by measuring 30 fields for each sample using a 20× lens.

Statistics. Data are expressed as mean ± SD. Statistical analysis was performed by two-sample Welch *t*-test (https://www.statskingdom.com/) to compare the results of experimental groups. A value of *p* < 0.05 was considered statistically significant.

## 3. Results

### 3.1. In Vitro Experiments

The neutral red viability assay demonstrated that the EAA-mix significantly reduced CT26 cell viability in both a dose- and time-dependent manner. Notably, after 72 h, there was a 60% decrease in cell survival. Interestingly, the administration of NEAA did not have any significant effect on cancer cell viability (Figure 1).

We aimed to assess the ability of the EAA-mix to trigger apoptosis in a cancer cell line. For this purpose, CT26 cells were exposed to the EAA-mix at concentrations of 0.5% or 1%. Using FACS analysis, we examined the apoptosis of cells treated with a 1% EAA-mix over a 96 h time course. Our findings revealed that the EAA-mix significantly escalated the proportion of apoptotic cells, demonstrating 60% of cell death after just 48 h of incubation and reaching 80% after 96 h (Figure 2A).

Furthermore, through immunofluorescence analysis for LC3β, we evaluated the EAA-mix’s ability to induce autophagy (AUT) and the formation of autophagosomes. After 24 h, CT26 cells treated with the EAA-mix displayed the presence of LC3β positive autophagosomes, achieving peak intensity after 48 h. In contrast, untreated cells exhibited a very faint LC3β signal (Figure 2B).

### 3.2. In Vivo Experiments

#### 3.2.1. Sub-Cutaneous (s.c.) CT26 Injection

The body weight (b.w.) of animals at the end of treatment showed a decrease of about 16% in StD and only 6.2% in EAARD-fed (Figure 3A). It was necessary to euthanize two StD-fed mice before the deadline by order of the veterinarian because they showed serious signs of suffering. No signs of suffering were observed in EAARD-fed mice, and all mice reached the deadline. There were no significant differences noted in the ratio of adipose tissue (rpWAT and BAT) to body weight in relation to diet (rpWAT: T = 1.09, *p*-value 0.291; BAT: T = 0.90, *p*-value 0.386) (Figure 3B). As such, the body weight loss observed in mice fed on a standard diet (StD) is primarily attributable to a reduction in lean mass. In fact, with StD a significant reduction in the muscle (triceps of surae) was observed in comparison to EAARD (T = 3.863; *p*-value 0.004) (Figure 3C).

#### 3.2.2. Subcutaneous Tumor Development

The progression of the subcutaneous (s.c.) tumor was monitored and measured at 7, 14, and 21 days post-injection of CT26 cells. Mice fed with the EAARD exhibited a significant slowdown in tumor development. Indeed, at the conclusion of the experiment (21 days), the average volume and weight of tumors in animals fed with the standard diet (StD) were over 6 times higher than those in animals on the EAARD. Specifically, the volume difference was statistically significant (T = 5.702; *p*-value < 0.001), as was the weight difference (T = 4.535; *p*-value < 0.001). Notably, StD-fed animals demonstrated a complex subcutaneous vascular network surrounding the tumor mass. This network was significantly reduced in animals on the EAARD (Figure 4).

### 3.3. Histopathological Examination

#### 3.3.1. Eosin/Hematoxylin (E/H) Staining

In the StD-fed samples, apoptotic cells were observed to be fewer and randomly dispersed, with the scarce presence of a mitotic cell (Figure 5A,B). Conversely, samples from EAARD-fed mice displayed a high number of foci containing apoptotic cell debris. These were characterized by cytoplasmic and nuclear condensation, as well as nuclear fragmentation. Mitotic cells were occasionally observed (Figure 5D,E).

#### 3.3.2. Sirius Red Staining

Under Sirius red staining at polarized light, thinner collagen fibers (type III) were uniformly distributed within the tumor from StD-fed mice, appearing yellow to green in color. In contrast, in the samples from EAARD-fed mice, collagen fibers were scarce and clustered within smaller areas (Figure 5C,F).

### 3.4. Immunohistochemical Analysis

#### 3.4.1. Anti-Ki-67 Staining

The magnitude of tumor cell proliferation was evaluated using anti-Ki-67, a recognized pathological proliferation marker in cancer cells. The nuclei were counted at 40x magnification. A total of 1524 nuclei were examined in StD-fed samples and 1650 in the EAARD-fed group. Nuclei were considered Ki-67-positive only when they exhibited intense staining. Tumors from StD-fed mice demonstrated a significantly higher percentage (8.14 ± 1.96) of intensely stained nuclei than those observed in EAARD-fed mice (1.83 ± 0.88) (T = 10.174; *p*-value < 0.001) (Figure 6A).

#### 3.4.2. Anti-GRP78 Staining

Given the key role of endoplasmic reticulum (ER) stress in abnormal protein folding, leading to cellular dysfunction and cell death, we examined changes in GRP78 staining. Tumor cells from StD-fed animals showed faint GRP78 staining. In contrast, in EAARD-fed animals, widespread and intense immunostaining was observed, particularly concentrated around the nuclei (T = 8.341; *p*-value < 0.001) (Figure 6B).

#### 3.4.3. Anti-Caspase-3 Staining

To evaluate the induction of apoptosis, we examined the staining for activated caspase-3. Tumors from StD-fed animals only occasionally showed cells with intense immunostaining. In contrast, the majority of tumor cells in EAARD-fed animals showed intense optical density of immunostaining (T = 15.029; *p*-value < 0.001) (Figure 6C).

#### 3.4.4. Anti-iNOS Staining

The presence of inflammation was evaluated by iNOS expression. Immunostaining was intensely expressed in all tumor cells from StD-fed animals. However, animals fed with EAARD showed significantly reduced staining, with only some cells intensely stained (T = 4.743; *p*-value < 0.001) (Figure 6D).

#### 3.4.5. Anti-CD31 Vascular Staining

To evaluate the extent of tumor mass vascularization, we measured the number of positive anti-CD31 vessels per unit area (Vascular density = Vessels/1000 µm^2^). The vascular density was higher in tumors from StD-fed animals compared to EAARD-fed animals (T = 3.692; *p*-value < 0.001) (Figure 6E). The total tumor surface analyzed in each group was about 150,000 µm^2^.

### 3.5. Intraperitoneal (i.p.) CT26 Injection

To evaluate any variations in the impact of different diets on in vivo tumor cell inoculation methods, we also performed i.p. injections on two groups of 15 animals each, fed with StD and EAARD, respectively. The goal was to assess the total number and volume of tumors in the abdominal cavity, distinguishable even to the naked eye.

Notably, only four mice in the StD-fed group reached the predetermined endpoint of the trial (21 days). Three animals died on days 12, 13, and 15 post-inoculation, and the remaining 8 had to be euthanized prior to the endpoint due to severe signs of distress, resulting in an average survival of 16 days post-injection (Figure 7A).

Conversely, nine mice in the EAARD-fed group reached the endpoint. One mouse died after 15 days and five had to be euthanized before the end of the trial. The average survival in this group was 19.5 days post-injection (Figure 7B).

All i.p. injected mice began to lose body weight between 5 and 10 days post-injection in both groups, continuing until the end of the observation period (Figure 7C). In both groups, the retroperitoneal white adipose tissue (rpWAT) was absent. Moreover, the brown adipose tissue (BAT) was significantly reduced in StD-fed animals (T = 2.909; *p*-value 0.01). Similarly, the triceps surae muscle mass was drastically reduced in the StD-fed animals. However, muscle mass was maintained in the EAARD-fed animals (T = 6.825; *p*-value < 0.001) (Figure 7D).

### 3.6. Mesenteric Tumors

Both experimental groups exhibited mesenteric tumors of varying numbers and sizes (Figure 8A,B). StD-fed animals presented a higher number of tumors and/or swollen lymph nodes (11.93 ± 4.33), both adjacent to the small intestinal wall and at the mesentery’s root. In contrast, EAARD-fed animals had a reduced number of tumors and mesenteric lymph nodes (7.07 ± 2.05) (Figure 8C,D). In both groups, some mice also had large, single intraperitoneal tumors resting on the abdomen’s posterior wall (insert in Figure 8C,D). These tumors were occasional and significantly smaller in EAARD-fed mice compared to those in StD-fed animals.

The total volume of tumors extracted from each group was significantly lower in EAARD-fed animals. The histopathological examination of E/H-stained sections revealed the same characteristics described for the two experimental groups in the subcutaneous (s.c.) inoculation (Figure 8E,F).

## 4. Discussion

The primary finding from this study’s data is that increasing the ratio of essential amino acids to nonessential amino acids (EAA/NEAA >> 1) significantly increases cell apoptosis and autophagy, therefore slowing down the growth of murine colon tumor cells (CT26) both in vitro and in vivo.

In vitro, tumor cells treated with EAA-mix exhibited a higher incidence of apoptosis. Similarly, in vivo experiments revealed high levels of nuclear chromatin thickening and fragmentation in tumors, along with strong immunostaining for activated caspase-3 in tumors from EAARD-fed mice. This aligns with previous studies showing that EAA-mix has a cell-dependent antiproliferative and cytotoxic effect, activating apoptotic pathways and leading to cancer cell death without affecting noncancer cells [7,20,26].

Apoptosis, a regulated and programmed cell death process, is crucial in tumor therapy [27]. Caspases, a family of proteolytic enzymes, are fundamental components of the apoptotic pathway [28]. Many anticancer therapies, including cytotoxic drugs, radiotherapy, and immunotherapy, can induce tumor cell death by activating caspase-3. Therefore, caspase-3 activation is often used as a surrogate marker for cancer treatment efficacy [29]. However, colon cancer patients with low levels of activated caspase-3 have been reported to have longer disease-free survival times [30]. Recent studies have suggested that caspase-3 may promote tumor growth by creating a proangiogenic microenvironment [31]. Despite this, caspase-3 knockout cells show impaired growth when seeded at low densities [29]. A recent study on the antitumor properties of *Cyclocaryapaliurus* polysaccharide (CP) on CT26 mouse colon carcinoma cells showed that CP induced cell apoptosis through improving caspase-3 activity, suggesting CP as a potential natural therapeutic agent for colon cancer [32].

In accordance with these findings, we observed a strong induction of activated caspase-3 following EAARD in s.c. tumors, correlating with the slower tumor growth rate in both s.c. and i.p. tumors. This was also confirmed in vitro by the significant increase in mortality of tumor cells treated with EAA-mix. This evidence suggests that caspase-3 activation induced by EAA promotes tumor cell death in vivo. Indeed, a previous study indicated that EAA-mix increased branched-chain amino acid oxidation and decreased glycolysis, ATP levels, redox potential, and intracellular content of selective NEAA in cancer cells. This led to elevated EAA/NEAA ratios, NEAA deficiency, and, consequently, NEAA starvation. The latter activated the stress pathway, mTOR inactivation, and apoptosis in cancer cells only [26].

In addition, our results showed that most tumor cells from StD-fed mice had numerous Ki-67-positive nuclei but not EAARD-fed mice. Ki-67 is a nuclear protein closely associated with cell proliferation [33,34] and is widely used to evaluate cell proliferation and aggressiveness in various malignant tumors [35]. High expression of Ki-67 suggests active proliferation and mitosis of tumor significantly associated with its histological differentiation [36]. Although data on Ki-67 in relation to chemotherapy in human colon carcinoma are sometimes conflicting, several studies showed that high Ki-67 expression was related with poor overall survival and may be used to predict patient prognosis [37]. Considering the literature, the decrease in Ki-67-positive nuclei observed in mice fed with EAARD suggests an impairment of cell replication potential, further supported by the smaller volume reached by the tumors, both resulting from s.c. or i.p. injection, and by proapoptotic markers.

Interestingly, tumors of animals fed with EAARD contained reduced vascularization. Tumor development is determined not only by the mitotic activity of its cells but also by the degree of development of the vascular bed and its capacity to fulfill all its functions [38]. Angiogenesis is the process by which new and abnormal intra-tumor blood vessels expand to accommodate tumor growth, metastasis, and metabolism promoting the anaerobic and glycolytic conditions that characterize the tumor microenvironment [39,40]. For these reasons, the inhibition of tumor angiogenesis is the target of many antineoplastic therapies [41]. We observed an intricate subcutaneous vascular network involving the tumor in StD-fed animals, while the vascular network was reduced in EAARD-fed mice. This difference in vascularization is maintained within the tumor, as demonstrated by anti-CD31 staining. Therefore, the poor vascularization observed in tumors of animals fed with EAARD, together with the slowing of tumor growth and the low presence of proinflammatory iNOS, suggests that excess of free EAA promotes maintenance of a microenvironment highly unfavorable for tumor cell metabolism and proliferation.

Another noteworthy morphological observation is the modulation of collagen syntheses in tumors of animals fed with EAARD. Collagen can directly promote and feed tumor growth [42]. Indeed, a recent study shows that solid tumor growth depends upon collagen binding and uptake mediated by the TEM8/ANTXR1 cell surface protein in tumor-associated stromal cells. These cells processed collagen into glutamine, which was then released and internalized by cancer cells. Under chronic nutrient starvation, a condition driven by the high metabolic demand of tumors, cancer cells exploited glutamine to survive [43]. So, to become malignant, epithelial cells need to acquire the ability to degrade molecules of extracellular matrix [44,45]. Because collagen can be a metabolic source to fuel cancer growth, the very scarce presence of newly formed collagen in tumors from EAARD-fed mice, suggests that tumor-associated stromal cells would lack substrates to convert into glutamine as an energy supplementary source, and this consequently would starve tumor cells. This fault may be a further strong drive activating either autophagy (AUT) and, finally, apoptosis.

Through the use of immunofluorescence, we have demonstrated that the EAA-mix appears to stimulate the induction of AUT in CT26 tumor cells. While this finding necessitates further detailed investigation, we believe it is a critical point that merits consideration. Indeed, AUT is an intracellular process that suppresses tumorigenesis by inhibiting cancer cell survival and inducing cell death, but, in contrast, it may also promote cancer cell survival, proliferation, and tumor growth. Mechanistically, those processes are controlled by a series of proteins, and peculiar focus has been established on mTORcs and the cascades controlling both mTOR activation and inhibition of AUT [46]. The observed increase in LC3β, which is a potential indicator of AUT, may suggest a “self-destructive” attempt by tumor cells to counterbalance the excess of EAA in order to produce the NEAA required to complete protein synthesis (which contain NEAA in large excess of EAA) and facilitate cell replication.

Evidently, these mechanisms are not sufficient, or excessively efficient, to protect cancer cells, which therefore undergo apoptosis and death. Indeed, ER-stress-associated AUT (termed ER-phagy) is essential for maintaining stabilized ER function via the degradation of aberrant unfolded protein and/or surplus components of the ER [47]. As we have already hypothesized [2], syntheses of all structures necessary to duplicate, both functional proteins and membrane lipids, require an enormous ATP to AMP consumption and, since AMP activates AMP kinase and this in turn further pushes AUT to perform, the sums of all those metabolic drives may rise limits of AUT towards activation of apoptosis. Current experiments are underway to determine whether the EAA-mix may play a role in the activation of the autophagic flux.

We also observed significant increases in ER stress, as evidenced by the increased expression of the 78 kDa glucose-regulated protein (GRP78) in tumors from EAARD-fed mice. GRP78, an ER chaperone, plays a central role in maintaining protein homeostasis across all cell types. GRP78, a member of the heat-shock protein family, is primarily located in the ER where it plays a key role in protein folding. Its expression is upregulated during the unfolded protein response (UPR), which is triggered in response to ER stress [48]. The secretion and translocation of GRP78 from the ER to the plasma membrane are associated with several pathological conditions, including autoimmune diseases [49], and tumors cells [50]. In these contexts, GRP78 plays a significant role in proliferation, angiogenesis, metastasis, and resistance to anticancer drugs [51]. In addition, it has been demonstrated that drug-induced expression of GRP78 prevents apoptosis and production of intracellular ROS, enhancing survival and proliferation of drug-resistant colon cancer cells via regulation of apoptosis-, survival-, and cell-cycle-associated signaling pathways [52]. Conversely, surface expression of GRP78 is an early response to inflammation and, consequently, over 90% of early apoptotic pancreatic cells express GRP78 on their cell membrane. This suggests that surface GRP78 acts as a proapoptotic receptor in pancreatic beta cells, with the underlying mechanism mediated by ER stress [53]. This difference in physiological response may depend on whether surface GRP78 is engaged at its N- or C-terminus. In cancer cells, ligation of surface GRP78 with blocking antibodies against the N-terminal domain induces proliferation [50], whereas ligation with blocking antibodies against the C-terminal domain promotes apoptosis [54]. Much evidence suggests that cellular dysfunction and cell death induced by ER stress are major contributors to many diseases. Therefore, modulators of ER stress pathways are potentially attractive targets for therapeutic discovery [55].

Our findings indicate that the EAARD diet, by altering EAA/NEAA, increased GRP78 and activated caspase-3, while reducing iNOS immunostaining in cancer cells. This suggests the presence of inflammation, deeply altered protein homeostasis inducing ER stress, and activation of apoptosis, which, in turn, impairs cell proliferation. These results are in line with those of Ragni et al., which showed that the specific EAARD diet, providing the introduction of altered EAA/NEAA, promoted ER stress and inhibited mTOR activity, thereby reducing tumor growth [26].

However, it is well known that EAA can enhance mTORC1 activity, promoting cell proliferation [9,14]. mTORC1 is activated by the availability of growth factors (mitogens, i.e., growth factors, extracellular signals capable of inducing cell duplication and proliferation, such as insulin), cellular energy (ATP), nutrients, and by some NEAA, such as arginine and glutamine. These stimuli trigger the synthesis of essential building blocks for the organism, such as proteins, lipids, and nucleotides, activating metabolic pathways that drive cellular and organismal growth [56]. Altered regulation of mTORC1 activity is closely associated with various diseases, including cancer, diabetes, and neurodegenerative disorders [57,58].

Considering these data, one might think that a large amount of EAA could also stimulate tumor cell proliferation. However, this is not necessarily the case. Only two experimental studies have reported that long-term leucine supplementation, a potent inducer of mTORC1, promoted the development of bladder cancer in rats [59,60]. However, from epidemiological data in humans affected by liver cirrhosis, chronic supplementation with BCAA formulations rich in leucine has been shown to significantly reduce the risk of liver cancer evolution [61]. Currently, there is insufficient evidence to establish a cause–effect relationship between leucine and cancer growth [26]. In fact, recent experimental studies suggest the opposite.

Studies have shown that a diet enriched in leucine (3%) can steer tumor metabolism towards a less glycolytic phenotype, resulting in a reduction in the Warburg effect. This is associated with decreased tumor aggressiveness and fewer metastatic sites [62]. Moreover, a recent literature review indicates that leucine, by activating mTORC1 and increasing protein synthesis, can decrease protein degradation, thereby mitigating the symptoms of cancer cachexia [63]. Furthermore, recent in vivo studies on tumor cells demonstrated that, after treatment with a complete mixture of EAA, the intracellular concentration of glutamate, glycine, aspartate, and alanine (all of which are NEAA) decreased, while that of EAA increased. This situation mimics a response to hunger, particularly a deficiency in glutamate, which activates the catabolism of BCAA at the mitochondrial level, inhibiting glycolysis (Warburg effect) in favor of the Krebs cycle. The decrease in intracellular NEAA levels, being a significant percentage of all amino acids needed for protein synthesis, ultimately activates ATF4, leading to reduced mTORC1 activity. The result is reduced tumor growth due to increased ER stress and reduced mTORC1 activity, which, in turn, supports autophagy. All these changes would eventually lead to increased apoptosis and cell death. Nontumor cells did not suffer these detrimental effects; indeed, their metabolism was actively supported by the EAA-mix [26]. Overall, these findings suggest that high levels of amino acids alone are not sufficient to increase tumor biomass. In order to serve as a metabolic fuel for cell proliferation, their supply must be linked to an unchallenged mTORC1 activation, as supported by any naturally occurring AA composition.

Cancer cells are highly dependent on an excess of NEAA for development, survival, and multiplication [18]. This is also evident in our in vitro data in which tumor cells proliferated normally with administration of NEAA alone. Serine, like glutamine and glycine, is an NEAA that tumors particularly crave and probably depend on to sustain multiple anabolic processes that support growth and proliferation [64]. Some cancer cells upregulate de novo serine synthesis [65,66,67], while many others depend on exogenous serine for optimal growth [15,68,69]. In fact, it has been proven that dietary restriction of serine and glycine can reduce tumor growth in xenograft and allograft models [15,70]. Additionally, in genetically engineered mouse models of lymphoma and intestinal tumors, a serine- and glycine-free diet increased survival compared to a control diet or normal diet that contains whole protein as a source of AA [17].

However, as previously noted, we believe that, by administering diets without serine and glycine and not substituting these amino acids with other NEEA, the authors did not consider that the EAA/NEAA ratio changes in favor of EAA. Therefore, it could be possible that the observed increase in survival does not depend on specific NEAA deficiency but on the proportional increase in EAA [18]. Interestingly, the EAA-mix used in our experiments contained serine but, despite this, we observed significant cancer cell death in vitro and a substantial slowdown of tumor progression in vivo, both subcutaneously and intraperitoneally.

Serine is commonly found in food proteins and can be easily synthesized from gluconeogenic AA. It can also be derived from glycine, but this reaction consumes NADH and depletes a specific methyl group of folates through this metabolic pathway. Notably, folate fortification of food has significantly reduced the incidence of colon cancer in the USA, according to a long-term epidemiological survey [71]. Our EAA-mix contained serine to achieve a 15% ratio of NEAA for three main reasons: (i) it is the quickest metabolically available amino acid for energy production and transamination, generating one pyruvate molecule; (ii) it plays a crucial role in maintaining folates charged with methyl groups [72]; and, (iii) in combination with EAA, it may have an anticancer effect as an allosteric activator of pyruvate kinase by forming a tetramer (isoform PKM2) [73]. This could potentially enhance the drive of glucose to full oxidation in mitochondria, allowing metabolic production of sufficient reactive oxygen species (ROS) to further activate the AUT pathway [20,74].

The results from this study indicate that an excess of EAA, leading to an increased EAA/NEAA ratio, can significantly and negatively influence colon cancer cell survival, exhibiting remarkable antiproliferative properties. In fact, if the EAA/NEAA ratio in the tumor cell is tilted in favor of EAA, the consumption of large quantities of ATP would be activated for protein synthesis, leading to an increase in ADP and AMP. The resulting low energy levels, due to the consumption of ATP for constructing the thousands of peptide bonds necessary for protein synthesis, activate AMPK, which, in turn, inhibits mTORC1 and activates autophagy, providing substrates to support new ATP production and a decrease in NEAA availability. These changes create a cycle that inhibits proliferation and induces apoptosis [2,18].

From a clinical perspective, these results challenge the common belief and suggest that the continuous administration of EAA in sufficient excess may represent a strategy to create an unfavorable metabolic environment for tumors, selectively impairing cancer metabolism and subsequently reducing cell growth, while maintaining muscle mass. Therefore, we may summarize our findings with the following statement: the administration of a special mix of EAA allows the patient to be nourished and the tumor to be starved.

### Study Limitation

A potential limitation of this study is the exclusive use of immunohistochemistry/immunofluorescence to support our in vivo results. However, as demonstrated in previous studies [75,76], this choice was made due to the heterogeneous cell population within tumors and the varying states of the cell cycle and differentiation among cancer cells. Consequently, histopathological changes, precise location of markers, and evaluation of their staining intensity can only be accurately determined with immunohistochemistry. In contrast, molecular analysis, while sensitive in detecting the presence of proteins, requires immediate freezing and homogenization of the sample and does not take into account the specificities of the protein’s location, tissue morphology, and organization. This is a major limitation in the exclusive use of molecular analysis, which we believe is comparable, if not superior, to immunohistochemistry. Therefore, we believe that our data, even if based on immunohistochemistry, are worth considering and form the basis for further studies. These data clearly highlight a slowdown in tumor progression due to the EAA-rich diet.

## 5. Conclusions

These data strongly suggest that dietary supplementation with an adequate amount of all free EAA is safe and promotes the slowdown of colon cancer cell proliferation in a mouse model. This effect could potentially be an effective integrated therapy to complement and support conventional therapies in cancer patients.

## Figures and Tables

**Figure 1 cells-13-01210-f001:**
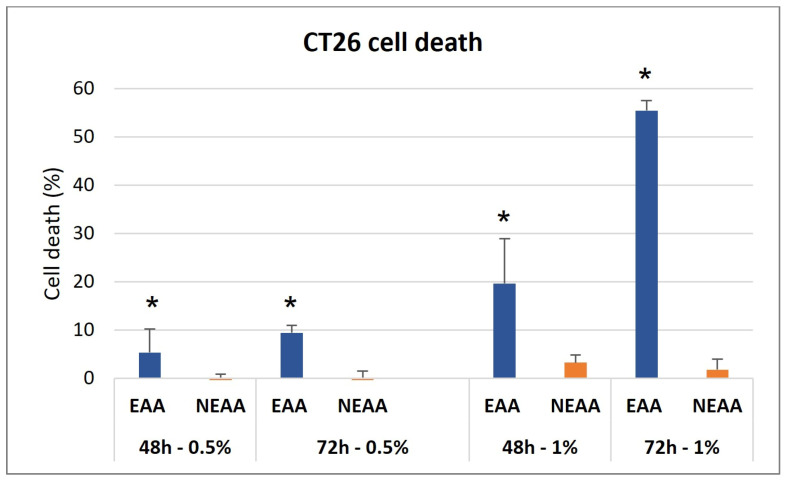
EAA-mix reduced colon carcinoma cell viability. Neutral red viability assay was performed on CT26 cells treated for 48 h and 72 h with EAA-mix or NEAA-mix at 0.5 and 1% concentration. Results are presented as percentage of cell death compared to untreated cells. * *p* > 0.001.

**Figure 2 cells-13-01210-f002:**
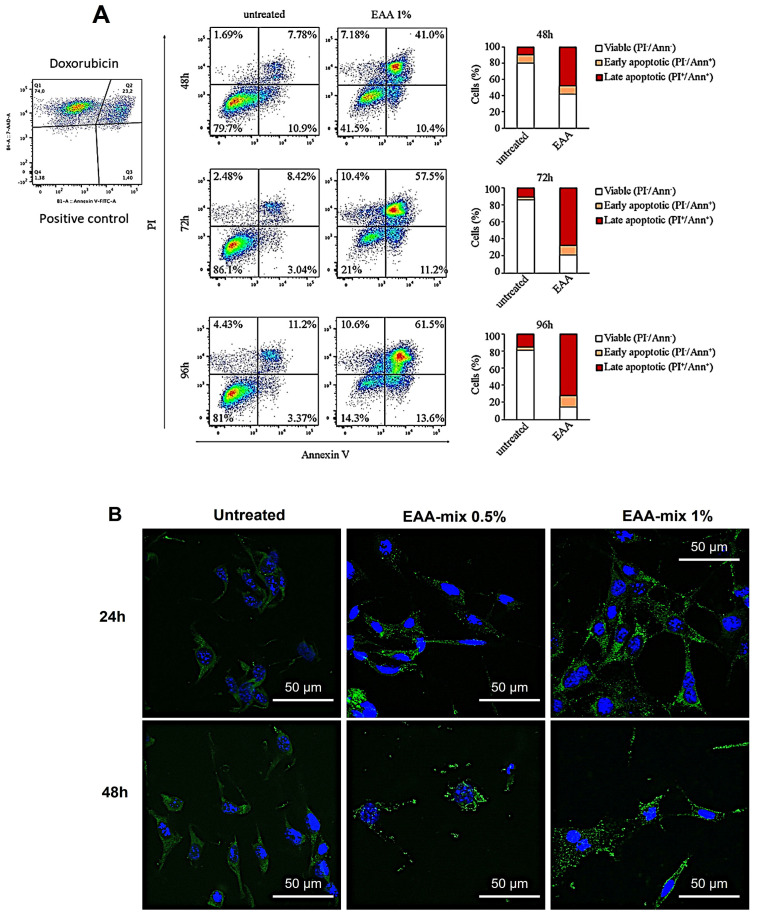
(**A**) Fluorescence-activated cell sorting (FACS) analysis was conducted to examine apoptosis in CT26 cells treated with 1% EAA at the specified time points. Doxorubicin was used as a positive control. (**B**) Immunofluorescence analysis for LC3β was carried out on CT26 cells that were incubated with the EAA-mix at concentrations of 0.5% and 1% for periods of 24 and 48 h. Representative images were captured using a 63x objective. The scale bar represents 50 µm.

**Figure 3 cells-13-01210-f003:**
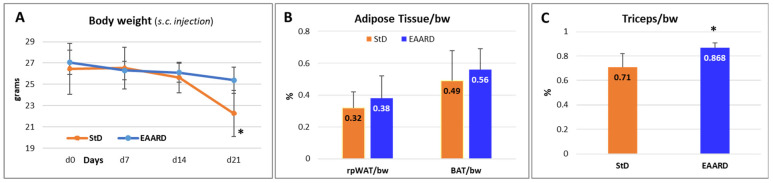
(**A**) Body weight (g) trend in the two experimental groups. (**B**) The ratio of rpWAT or BAT to body weight did not show a change between StD-fed and EAARD-fed mice. (**C**) Triceps of surae muscle weight was maintained in mice fed with the EAARD. Values are expressed as mean ± sd. * *p* < 0.05.

**Figure 4 cells-13-01210-f004:**
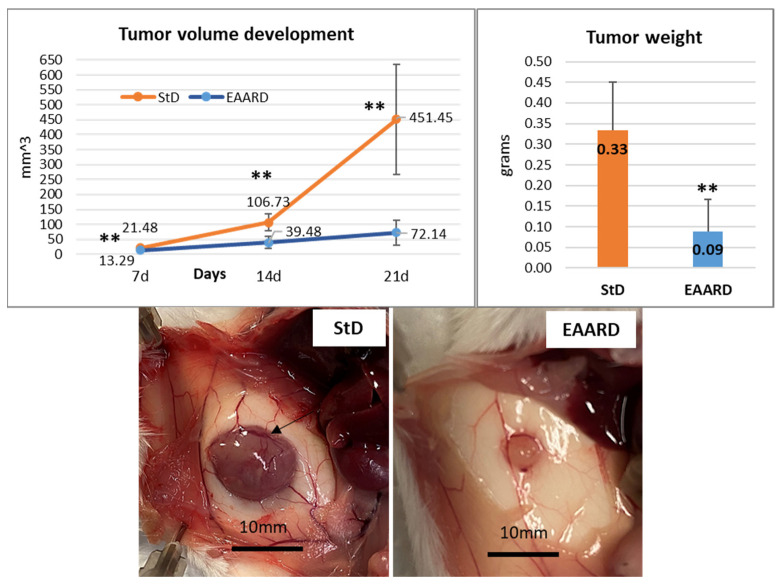
Above. Timing of tumor volume development (mean ± sd) (left) and mean tumor weight (mean ± sd) according to diets (right). Below. Representative s.c. tumor in StD-fed and EAARD-fed mice. In StD-fed animals, note the presence of an intricate subcutaneous vascular network that also involves the tumor (arrow). Scale bar: 10 mm. ** *p* < 0.01.

**Figure 5 cells-13-01210-f005:**
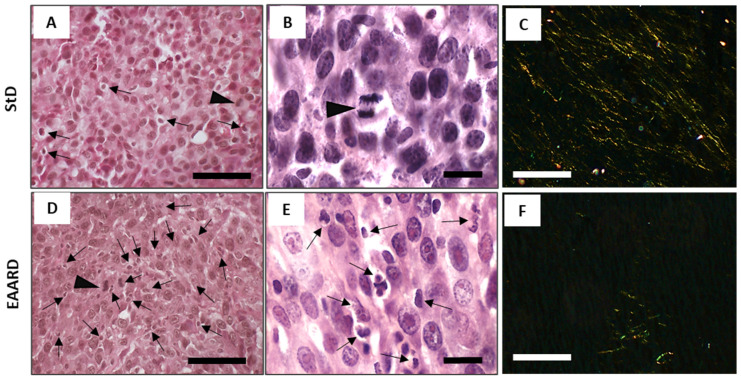
S.c. injection. Representative tumor H/E and Sirius red (polarized light) stained sections in StD-fed (**A**–**C**) and EAARD-fed mice (**D**–**F**). Cytoplasmic and nuclear condensation and fragmentation, and foci of apoptotic cell debris are found (black arrows) in StD-fed and, much more, in EAARD-fed mice. Mitotic cells (arrowhead) were very scarce in both diets. Note the presence of abundant thin stromal collagen fibers in tumors with StD (**C**). In those with EAARD, the presence of stromal collagen is very scarce and tends to be concentrated in restricted areas (**F**). Scale bar: (**A**,**C**,**D**,**F**) 100 µm; (**B**,**E**), 20 µm.

**Figure 6 cells-13-01210-f006:**
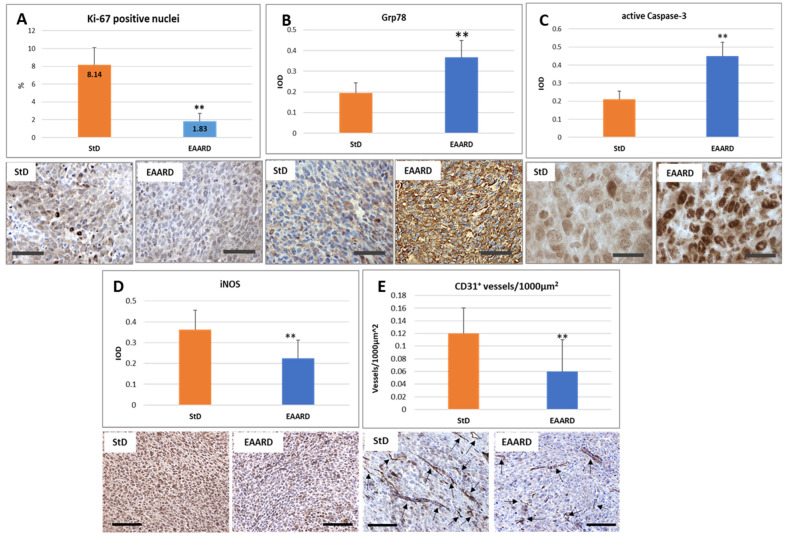
S.c. injection. Immunohistochemistry data and representative images of tumor tissue immunostaining. (**A**) Ki-67 is expressed as percentage of intense immunostained nuclei in each section. StD-fed tumors show more cells intensely stained. Scale bar: 50 µm. (**B**–**D**) The graphs indicate the optical density for GRP78, active-Caspase-3, and iNOS, which appears to be significantly different in tumors from animals fed EAARD. Scale bar: (**A**,**B**,**D,E**), 50 µm; (**C**), 20 µm. (**E**) CD31 staining is expressed as number of vessels for unit area. EAARD determines the significant reduction in the vascularization of the tumor. Scale bar: 50 µm. ** *p* < 0.01.

**Figure 7 cells-13-01210-f007:**
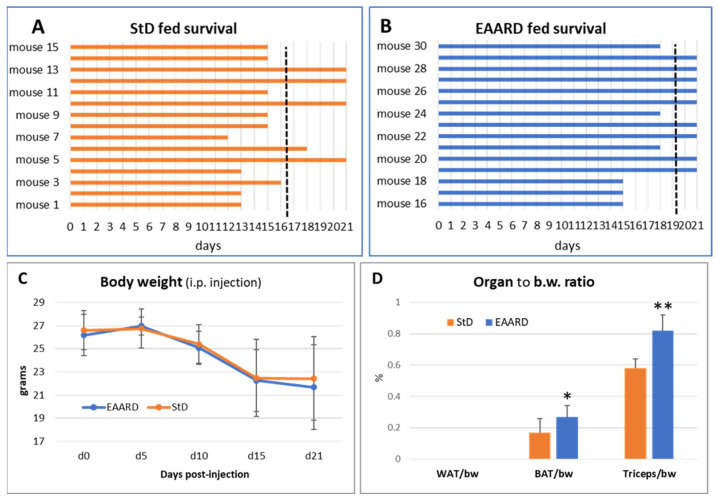
I.p. injection. (**A**,**B**) Survival time between the day of inoculation (day 0) and the day of sacrifice at the end of the planned observation period (day 21). The dashed black line indicates the average survival days. The mean survival time for StD-fed and EAARD-fed was 16.5 and 19.5 days, respectively. (**C**) Mice body weight change after i.p. CT26 inoculation (mean ± sd). (**D**) Ratio between organ weight and body weight according to group (mean ± sd). * *p* < 0.05; ** *p* < 0.01.

**Figure 8 cells-13-01210-f008:**
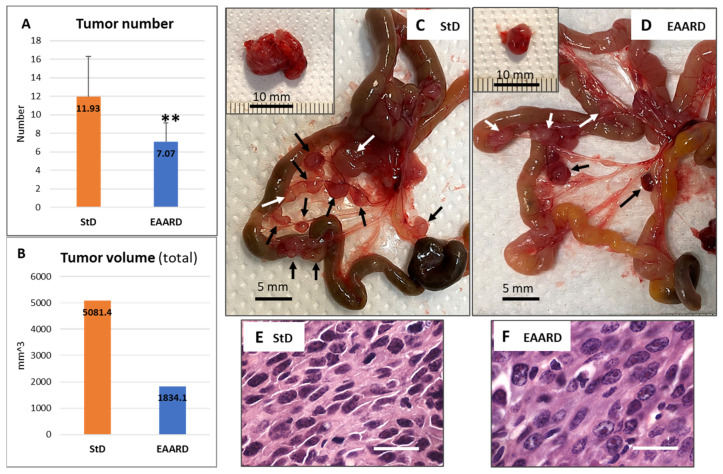
i.p. injection. (**A**) Abdominal number of tumors clearly distinguishable even to the naked eye (mean ± sd). ** *p* < 0.01. (**B**) Total tumor volume according to group. (**C**,**D**) Representative image of the mesenteric tumor (arrows black and white) in StD-fed and EAARD-fed, respectively, after 21 days from injection. Scale bar: 5 mm. (**C**,**D**) insert. Large intraperitoneal tumor sometimes observed in animals fed with StD and EAARD. Scale bar: 10 mm. (**E**,**F**) Representative E/H-stained sections. Nuclear condensation and fragmentation and foci of apoptotic cell debris are found mostly in EAARD-fed mice. Scale bar: 20 µm.

**Table 1 cells-13-01210-t001:** Composition of EAA-mix and NEAA-mix for in vitro experiments, and composition of pellets (StD and EAARD) for mice. * Nitrogen (%) from free AAs only. ° Nitrogen (%) from vegetable and animal proteins and added AA. StD = standard diet; EAARD = essential-AA-rich diet; N = nitrogen. bcaa = branched chain AA.

	EAA-mix	NEAA-mix	StD	EAARD
KCal/Kg	--	--	3952	3995
Carbohydrates (%)	--	--	54.61	61.76
Lipids (%)	--	--	7.5	6.12
Nitrogen (%)	--	--	21.8 °	20 *****
Proteins: % of total N content	--	--	95.93	--
Free AA: % of total N content	--	--	4.07	100
EAA/NEAA (% in grams)	--	--	< or <<0.9	6.14 (86/14)
Free AA composition (%)				
L-Leucine (bcaa)	13.53	--	--	13.53
L-Isoleucine (bcaa)	9.65	--	--	9.65
L-Valine (bcaa)	9.65	--	--	9.65
L-Lysine	11.6	--	0.97	11.6
L-Threonine	8.7	--	--	8.7
L-Histidine	11.6	--	--	11.6
L-Phenylalanine	7.73	--	--	7.73
L-Methionine	4.35	--	0.45	4.35
L-Tyrosine	5.80	1.0	--	5.80
L-Tryptophan	3.38	--	0.28	3.38
L-Cystine/Cysteine	8.20	--	0.39	8.20
L-Alanine	--	35.0	--	--
L-Glycine	--	15.0	0.88	--
L-Arginine	--	14.0	1.1	--
L-Proline	--	12.0	--	--
L-Glutamine	--	12.0	--	--
L-Serine	2.42	6.0	--	2.42
L-Glutamic Acid	--	2.0	--	--
L-Asparagine	--	2.0	--	--
L-Aspartic Acid	--	1.0	--	--
Ornithine-αKG	2.42	--	--	2.42
N-acetylcysteine	0.97	--	--	0.97

## Data Availability

The data are available on request from the corresponding author.

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
