# Peer review of "Intake of Special Amino Acids Mixture Leads to Blunted Murine Colon Cancer Growth In Vitro and In Vivo"

_cells, 2024, doi:10.3390/cells13141210_

Round 1

Reviewer 1 Report (Previous Reviewer 2)

Comments and Suggestions for Authors

The paper is now suitable for publication in Cells

Author Response

We are grateful to this referee for taking the time to review. 

Reviewer 2 Report (Previous Reviewer 3)

Comments and Suggestions for Authors

The authors of the manuscript no. cells-3027130 have addressed convincingly the points that were raised in the review. The corrections introduced into the manuscript improve its quality.  It can be accepted in this form.

Author Response

We are grateful to this referee for taking the time to review. 

Reviewer 3 Report (New Reviewer)

Comments and Suggestions for Authors

1. A similar review was carried out in Turnitin, obtaining 13%, it is recommended to verify the origin.

2. The publication is correct in the background and the results are consistent with the objective of the research. I have no observations directly related to the research and publication of the article.

Author Response

Comment 1. A similar review was carried out in Turnitin, obtaining 13%, it is recommended to verify the origin.

Answer. We are grateful to this referee for taking the time to review. We believe that 13% overlap detected in Turning is due to the preprint (indicated below and never published) of a brief communication of preliminary data, which was subsequently expanded and integrated with other experiments described in this work submitted to Cells.

Preprint: Corsetti G et al. “Preliminary Data: Feeding and Colon Cancer Growth. Modified Nitrogen Intake Leads to Blunted Cancer Growth Independently by Serine”. April 2023. DOI: 10.20944/preprints202304.0068.v. License CC BY 4.0

We carefully checked for any overlaps detected in Turnitin. Almost all of them refer to the technical protocols of the methods of our previous works. However, we have changed some parts in the discussion (in blue). 

This manuscript is a resubmission of an earlier submission. The following is a list of the peer review reports and author responses from that submission.

Round 1

Reviewer 1 Report

Comments and Suggestions for Authors

-There is flaw in methodology.  The Composition of EAA and NEAA in table 1 in not accurate as EAA mixture contain non EAA.

- What is bioavailability and interaction???

Comments on the Quality of English Language

Extensive English editing

Reviewer 2 Report

Comments and Suggestions for Authors

The paper entitled “Intake of special amino acids mixture leads to blunted murine 2 colon cancer growth in vitro and in vivo” by Corsetti et al  investigates wether EAA administration can influence cancer survival and may improve the nutritional status. The study performed poth in vitro and in vivo, appears well conducted and the paper is well written, however some aspects should be considered to improve the scientific soundness of the work.

aa) A major concern regards  the number of dead mice upon the intraperitoneal injection of CT26 cells ad under the standard diet. 11 mice of the 15 treated did not reach the predetermined endpoint of the trial, can the authors explain this mortality rate? This of course highlights the better conditions and survival of EAARD fed mice  but may raise some concerns regarding the analysis of the state of nutrition of the survivors. For example in figure 7D the comparison of  BAT/bw ratio between StD and EAARD fed mice appears not significant when considering the standard deviation.

 b) Considering Fig 5, authors claim that in mice treated with subcutaneous injection of CT26 cells and EAARD fed, mitotic cells can be less frequently observed, but in the figure 5D a mitotic cell is shown as in figure 5A. I suggest the authors to perform a statistical  analisys to show significant differences regarding this aspect.

 c)  In Figure 3B the reduction in rpWAT and BAT in StD fed mice seems not so clear considering standard deviations. How te reduction is calculated? It should be better to show the ratio between the adipose tissue and the body weight.

 d) At pag 13, line 374 an incorrect reference (27) is indicated.

Reviewer 3 Report

Comments and Suggestions for Authors

In their manuscript Corsetti et al. investigated the effect special amino acids mixture on murine colon cancer growth in in vitro and in vivo model. The authors revealed that increasing the ratio of essential amino acids to non-essential amino acids (EAA/NEAA >>1) causes cell apoptosis and autophagy, thus slowing down the growth of colon tumor cells in both in vitro and in vivo condition.

The experimental design has been properly planned and results have been well documented.

I have only minorobservations:

1.      Line 37, the term "mutated cells" can be replaced with "abnormal cells".

2.       In the Introduction section can be define term "essential and non-essential amino acids are and name these molecules.

3.       In the manuscript body p-value<0.000 should be improved to <0.001.